# Aequorin as a Useful Calcium-Sensing Reporter in *Candida albicans*

**DOI:** 10.3390/jof7040319

**Published:** 2021-04-20

**Authors:** Dominique Sanglard

**Affiliations:** Institute of Microbiology, University of Lausanne and University Hospital, CH-1011 Lausanne, Switzerland; Dominique.Sanglard@chuv.ch; Tel.: +41-21-3144083

**Keywords:** *Candida*, aequorin, calcium homeostasis, alkaline stress, amiodarone

## Abstract

In *Candida albicans*, calcium ions (Ca^2+^) regulate the activity of several signaling pathways, especially the calcineurin signaling pathway. Ca^2+^ homeostasis is also important for cell polarization, hyphal extension, and plays a role in contact sensing. It is therefore important to obtain accurate tools with which Ca^2+^ homeostasis can be addressed in this fungal pathogen. Aequorin from *Aequorea victoria* has been used in eukaryotic cells for detecting intracellular Ca^2+^. A codon-adapted aequorin Ca^2+^-sensing expression system was therefore designed for probing cytosolic Ca^2+^ flux in *C. albicans*. The availability of a novel water-soluble formulation of coelenterazine, which is required as a co-factor, made it possible to measure bioluminescence as a readout of intracellular Ca^2+^ levels in *C. albicans*. Alkaline stress resulted in an immediate influx of Ca^2+^ from the extracellular medium. This increase was exacerbated in a mutant lacking the vacuolar Ca^2+^ transporter *VCX1*, thus confirming its role in Ca^2+^ homeostasis. Using mutants in components of a principal Ca^2+^ channel (*MID1*, *CCH1*), the alkaline-dependent Ca^2+^ spike was greatly reduced, thus highlighting the crucial role of this channel complex in Ca^2+^ uptake and homeostasis. Exposure to the antiarrhythmic drug amiodarone, known to perturb Ca^2+^ trafficking, resulted in increased cytoplasmic Ca^2+^ within seconds that was abrogated by the chelation of Ca^2+^ in the external medium. Ca^2+^ import was also dependent on the Cch1/Mid1 Ca^2+^ channel in amiodarone-exposed cells. In conclusion, the aequorin Ca^2+^ sensing reporter developed here is an adequate tool with which Ca^2+^ homeostasis can be investigated in *C. albicans*.

## 1. Introduction

Fungal pathogens need to adapt rapidly to the different environments encountered in their host and need signaling molecules for this purpose. Among them, calcium ions (Ca^2+^) play an important role. Changes in Ca^2+^ intracellular concentration in fungi mediate the rapid activation of signaling pathways linked to diverse cellular processes [1]. Ca^2+^ intracellular concentration therefore needs to be strictly controlled. In fungi, the cytosolic calcium concentration, [Ca^2+^], is generally maintained at the sub-micromolar level by a calcium homeostasis system [2]. Ca^2+^ channels located in the plasma membrane and/or internal compartments can open to cause a rapid increase in cytosolic [Ca^2+^] in a stimulus-dependent manner. This is usually followed by a return to normal cytosolic [Ca^2+^] through efflux from the cell or by uptake into specific organelles that serve as internal stores (vacuoles, mitochondria, endoplasmic reticulum) [2]. *Candida albicans* is one of the major fungal pathogens affecting human health [3]. *C. albicans* uses [Ca^2+^] flux in order to rapidly modulate responses important for morphogenesis [4,5]. For example, Brand et al. [4] described that contact sensing and subsequent hyphal development is strongly affected by extracellular calcium chelation. [Ca^2+^] flux is also important for *C. albicans* to cope with different stresses, and in particular with antifungal-dependent stresses [6,7,8]. One of the major stress response pathways in *C. albicans* is the calcium-dependent calcineurin pathway [9]. Blocking this pathway by chemical or genetic interference leads to a loss of resistance to these stresses [10]. In *C. albicans*, the calcium-dependent calcineurin pathway is also associated with the maintenance of virulence and thus highlights the importance of [Ca^2+^] flux control in the context of the host [7]. In this fungal pathogen, Ca^2+^ entry has been showed to be mediated at least by a channel complex composed of *CCH1* and *MID1* [11]. *C. albicans* harbors a putative Ca^2+^/H^+^ exchanger, Vcx1, a Ca^2+^ pump, Pmc1 [12], and a Ca^2+^ channel, Yvc1, in the vacuolar membrane [13]. Interestingly, Yvc1 in *C. albicans* was shown to play a role in cell polarization and hyphal extension, which is critical for the maintenance of virulence in this pathogenic yeast species [14]. A Ca^2+^/Mn^2+^ pump, Pmr1, has been described in the Golgi membrane [15], which may collaborate to maintain cytosolic Ca^2+^ homeostasis. In *Saccharomyces cerevisiae*, its deletion increases cytosolic [Ca^2+^] [16].

One issue in the measurement of Ca^2+^ flux in fungi is the detection of intracellular [Ca^2+^]. Several indirect reporters of intracellular [Ca^2+^] have been used, including fluorescent dyes and Ca^2+^-binding proteins that emit luminescence or are linked to the reconstitution of GFP [17]. Here, we report the use of a codon-adapted form of aequorin in *C. albicans* for the measurement of intracellular Ca^2+^. Aequorin is a 22 kDa calcium-activated photoprotein isolated from the hydrozoan *Aequorea victoria* [18]. The aequorin photoprotein is composed of apoaequorin, coelenterazine and bound oxygen, and shows a high affinity for free Ca^2+^. On binding to Ca^2+^, aequorin is converted into apoaequorin, carbon dioxide and coelenteramide, and energy from this reaction is released as blue light. The amount of released luminescence is dependent upon the concentration of free Ca^2+^, and thus aequorin can be used to measure intracellular [Ca^2+^] [19]. In this study, the use of a novel water-soluble coelenterazine formulation to reconstitute a functional light-emitting protein enabled light emission in whole *C. albicans* cells after exposure to various extracellular cues. This Ca^2+^-reporter system was used to probe Ca^2+^ flux in mutants lacking key Ca^2+^ exchangers/channels in this important fungal pathogen.

## 2. Materials and Methods

### 2.1. Strains and Media

Strains used in this study are listed in Appendix A. Strains were grown in the following media (when grown on solid media, 2% agar (Difco) was added): complete medium yeast extract peptone dextrose (YEPD): 1% Bacto peptone (Difco Laboratories, Basel, Switzerland), 0.5% Yeast extract (Difco) and 2% glucose (Fluka, Buchs, Switzerland); YEPD supplemented with 200 µg/mL nourseothricin (Werner BioAgents, Jena, Germany); minimal medium yeast nitrogen base (YNB): YNB (Difco) and 2% glucose (Fluka); YNB supplemented with complete supplement mixture (CSM) (MP Biomedicals); YNB supplemented with CSM without uracil (−ura) (MP Biomedicals, Luzern, Switzerland); FOA agar: YNB supplemented with 50 mg/mL uridine and 0.34 mg/mL FOA (5-fluoroorotic acid). *E. coli* DH5a was used as a host for plasmid constructions and propagation. DH5a was grown in LB broth or LB-agar plates, supplemented with ampicillin (0.1 mg/mL) or chloramphenicol (0.1 mg/mL) as required.

### 2.2. Aequorin Expression in C. albicans

The aequorin gene in the yeast expression vector pEVP11 [20] was codon-optimized for expression for *C. albicans*. CTG codons in the aequorin ORF (codons 96 and 133) were first replaced by TTG codons in two rounds of fusion PCRs. A first fusion PCR used 2 fragments originating from PCR products obtained with primer pairs AOQ-5/AOQ-NHE and AOQ-3/AOQ-PST (Appendix A) using pEVP11 as a template. A second fusion PCR used 2 fragments originating from PCR products obtained with primer pairs AOQ2-5/AOQ-NHE and AOQ2-3/AOQ-PST (Appendix A), using the first fusion product as a template. The *C. albicans* aequorin expression plasmid was constructed as follows: First, the *ACT1* promoter from pACT1-GFP [21] was cloned as a 1 kb XhoI-PstI fragment into the same sites of pGFP [21] to give pDS1208. The codon-optimized aequorin gene was amplified in the second fusion step as described. After digestion by PstI and NheI, it was cloned into the vector backbone of pDS1208 after digestion with the same restriction enzymes, to replace GFP to give plasmid pDS1209 (Appendix A), in which the expression of aequorin is driven by the *ACT1* promoter. The *URA3* marker in pDS1209 was replaced by the *SAT1* dominant marker as a 1.8 kb NotI-SpeI fragment from pVT50 (Sanglard lab collection) in the NotI/XbaI-digested vector background to give pDS1885 (Appendix A). Both plasmids were digested with StuI for integration into the *RPS1* locus of *C. albicans* [22].

### 2.3. Luminescence Measurements

*C. albicans* strains were grown overnight in YNB−ura liquid medium at 30 °C under constant shaking. Cells were next diluted to a cell density of 2 × 10^7^ cells/mL in 10 mL YNB−ura and grown under the same condition for 2 h. Water-soluble coelenterazine (catalog #3032, Nanolight Technology, Pinetop, AZ, USA) was resuspended in 5 mL PBS (phosphate buffered saline: 137 mM NaCl, 2.7 mM KCl, 4.3 mM Na_2_HPO_4_, 1.47 mM KH_2_PO_4_) and kept in the dark. An aliquot of 100 µL of this suspension was added to the 10 mL culture, wrapped with aluminium paper, and incubated for a further 3 h. The cells were next kept on ice, centrifuged, washed with cold 1% glucose water solution, and finally resuspended in 1–1.5 mL 1% glucose water solution with 5 mM Na-acetate (pH 5.0; GLAC medium) to a cell density of 2–4 × 10^8^ cells/mL.

Luminescence was read in a FLUOstar Omega instrument (BMG Labtech, Champigny/Marne, France). Cells (100 µL) were dispensed in a microtiter plate (OptiPlate-96 plate, PerkinElmer, Schwerzenbach, Switzerland). The instrument was kept at a temperature of 30 °C. Baseline luminescence was determined by shaking the plate for 3 s and reading the luminescence at intervals of 1.5 s for 10–20 s. This was followed by the injection of small volumes (5–15 µL) of diverse reagents followed by 1 s of shaking. Luminescence was read for another 110–120 s with 1.5 s intervals. To quantify total luminescence emitted by each strain, SDS and CaCl_2_ were injected to final concentrations of 0.15% and 100 mM, respectively. This aimed to permeabilize cells in order to measure total possible emitted luminescence for each cell type investigated in this study. Total possible luminescence was calculated by area under the curve (AUC) and were always above AUCs calculated in each of the stress experiments. Alkaline stress was induced by the simultaneous injection of 15 µL 0.25 M Tris-HCl (pH 7.5) and 10 µL 1 M CaCl_2_ to the cell suspension, which resulted in a pH shift from 5.0 to 7.5. Amiodarone hydrochloride (Sigma-Aldrich Chemie GmbH, Buchs, Switzerland) was prepared from a frozen stock of 50 mM DMSO by first diluting it to 25 mM with EtOH with further dilution to 0.5 mM in GLAC medium with 10% EtOH. An aliquot of 10 µL of this suspension was injected onto cells (50 µM final concentration). Luminescence data were exported to GraphPad Prism software (Version 9.1.0, GraphPad Software, San Diego, CA, USA) for data quantification and representation.

### 2.4. Mutant Construction

The *VCX1* deletion mutant was constructed using a URA-blaster method. *VCX1* flanking regions (around 500-bp) were amplified by primer pairs VCX-Kpn/VCX-BgB and VCX-PstB/VCX-HinB (Appendix A) and cloned sequentially into compatible sites of pMB7 [23] to obtain plasmid pDS1629. The plasmid was linearized with ApaI and SacI to liberate the deletion cassette that was used in *C. albicans* transformations. After deletion of the first *VCX1* allele and verification by PCR, the *URA3* marker was recycled by 5-FOA resistance selection on FOA agar resulting in isolate DSY4026. A second round of deletion was carried out in this isolate with the same deletion cassette to obtain the homozygous *VCX1* mutant DSY4034. The *URA3* marker was removed through 5-FOA-resistance selection to obtain the Ura− isolate, DSY4058.

### 2.5. Yeast Transformation

Overnight precultures of *C. albicans* strains in YEPD were diluted 50-fold in 50 mL fresh medium and incubated at 30 °C to a cell density of 2 × 10^7^ cells/mL. Cultures were washed twice in LiAc/TE (TE: 10 mM Tris-HCl pH 7.5, 1 mM EDTA; 0.1 M LiAc pH 7.5) and resuspended in 200 µL of LiAc/TE. Competent cells were stored at 4 °C. Competent cells (50 µL) were mixed with 5 µL denatured salmon sperm DNA (10 mg/mL), 5–10 µg of plasmids of interest, and 300 µL PEG (TE with 0.1 M LiAC pH 7.5, 40% PEG 4000). Cells were incubated for 30 min at 30 °C then 15 min at 42 °C. Transformed cells were pelleted and washed in TE and plated onto selective media. Plates were incubated onto the corresponding selective medium for 2 days at 34 °C.

### 2.6. Immunodetection of Aequorin

Proteins were extracted from log-phase cells using alkaline lysis according to [23]. Protein extracts were separated by SDS—10% polyacrylamide gel electrophoresis (PAGE) and transferred by Western blotting to a nitrocellulose membrane. Pre-stained molecular weight proteins (Biorad) were used as standards. Congo red staining of blotted proteins was carried out as described elsewhere [24]. Immunodetection of aequorin was performed with an aequorin polyclonal antibody (Covalab SAS, Bron, France). Signals were revealed by chemoluminescence (ECL kit, GE Healthcare, Glatbrugg, Switzerland) with anti-rabbit HRP-labeled secondary antibody. Quantifications of Western blot signals were carried out by an ImageQuant LAS 4000 System (GE Healthcare Life Sciences, Glattbrugg, Switzerland). Chemoluminescence signals and Congo red staining intensities were quantified by the software ImageQuantTL (Version 1.3).

### 2.7. Statistics

Comparisons of RLU values between conditions were evaluated using GraphPad Prism (version 9.1.0, GraphPad Software, San Diego, CA, USA) by 2-way ANOVA tests comparing each time point for each condition with Fisher’s least significant difference (LSD) method.

## 3. Results

### 3.1. Establishing Aequorin Expression in C. albicans

A codon-optimized version of aequorin for *C. albicans* was designed based on the aequorin gene available on the yeast expression vector pEVP11 [20]. The modified aequorin gene was placed under the control of the *ACT1* promoter to obtain pDS1209 and pDS1885, plasmids containing the *URA3* complementation marker or the dominant marker *SAT1*, respectively (Appendix A). The expression of aequorin was verified by immuno-blot analysis in wild-type and mutant *C. albicans* strains used in this study and few variations were observed (Figure 1A). The first attempts to obtain luminescence signals with whole *C. albicans* cells incubated with DMSO-soluble coelenterazine yielded results of variable quality, with low luminescence signals upon cell permeabilization (see below). This may have been caused by the poor penetration of coelenterazine in *C. albicans*; therefore, a novel water-soluble coelenterazine formulation [25] was used instead. After incubation with this formulation, high luminescence signals were obtained when cells were permeabilized with 0.15% SDS in the presence of 100 mM CaCl_2_ (Figure 1B).

### 3.2. Probing of Ca^2+^ Flux by Alkaline Stress

Using these experimental conditions, several Ca^2+^-mobilizing stimuli were tested with wild-type *C. albicans* cells and mutants lacking specific Ca^2+^ channels. It is known that shift to alkaline pH (alkaline stress) can result in increased cytoplasmic [Ca^2+^] in *C. albicans* [26]. When cells were exposed to a Ca^2+^ stress only (100 mM CaCl_2_), an increase in luminescence was observed in the wild-type, but at reduced levels (Figure 2A, solid blue line). When the pH of the medium was shifted by a pulse of a buffer from an acidic to an alkaline value (pH 5–7.5) together with 100 mM CaCl_2_, the aequorin-based Ca^2+^ capture system detected a significant increase in luminescence within seconds (Figure 2B). The pH shift produced a 20-fold increase in total luminescence in the wild-type as compared to CaCl_2_-only conditions.

Mutants lacking the *CCH1* and *MID1* channels were tested in the same alkaline-induced conditions. The data showed that single *cch1*∆/∆ and *mid1*∆/∆ mutants exhibited significantly reduced Ca^2+^ accumulation as compared to wild-type, while the combination of both *CCH1* and *MID1* deletion resulted in a further reduction in Ca^2+^ accumulation (Figure 2B). The same trend was observed when only Ca^2+^ stress (100 mM) was added; however, luminescence signals were also much reduced as compared to alkaline stress conditions (Figure 2A,B). Taken together, these data are consistent with the idea that alkaline stress can trigger a rapid Ca^2+^ accumulation in *C. albicans*, as previously published [26]. These data are also consistent with the requirement for the Cch1/Mid1 channel complex to mediate calcium entry into the cell cytoplasm in *C. albicans* [26,27]. Our data also suggest that additional Ca^2+^ uptake should be present in *C. albicans* because the absence of the Cch1/Mid1 channel complex could not completely abolish the activity of the aequorin reporter system (Figure 2B). One likely additional candidate in this process could be Factor Induced Gene 1 (Fig1), a component of the low-affinity calcium-uptake system (LACS), which is activated under specific conditions in *C. albicans* [28].

Vcx1 is a H^+^/Ca^2+^ exchanger located in the vacuolar membrane and uses the proton gradient across the vacuolar membrane (generated by a V-ATPase) to drive Ca^2+^ transport. Vcx1 enables excess cytoplasmic Ca^2+^ to be stored in the vacuole [29]. In the light of the previous results whereby alkaline stress induced a rise in cytoplasmic [Ca^2+^], it was expected that, in the absence of Vcx1, this excess of Ca^2+^ would be further exaggerated. Indeed, the *vcx1*∆/∆ mutant emitted about a 2.5-fold higher luminescence than the wild-type (Figure 3). Interestingly, the luminescence emission profile of the *vcx1*∆/∆ mutant exhibited a delay in the re-establishment of basal luminescence, indicating a perturbation in the kinetics of Ca^2+^ flux. These data are consistent with a reduced capacity of the *vcx1*∆/∆ mutant to store excess Ca^2+^ in the vacuole. Thus, the aequorin-based Ca^2+^ reporter assay enabled the verification of the role of Vcx1 in the maintenance of Ca^2+^ homeostasis in *C. albicans*.

### 3.3. Probing of Ca^2+^ Homeostasis by Amiodarone-Induced Stress

Next, the effect of the antiarrhythmic drug amiodarone on Ca^2+^ homeostasis was tested with the wild-type *C. albicans* and a mutant lacking both *CCH1* and *MID1*. Amiodarone induces Ca^2+^ spikes in the budding yeast *S. cerevisiae* [30] and is also known to have fungicidal effects in *C. albicans* [31]. *C. albicans* cells were exposed to a low dose of amiodarone (50 µM) and aequorin-dependent luminescence was detected even in the absence of external CaCl_2_ sources (Figure 4A, closed circles). Within seconds after amiodarone exposure, luminescence was emitted. This signal suggests either that trace Ca^2+^ was present in the testing medium (GLAC) and/or that internal Ca^2+^ stores were mobilized upon amiodarone stress. The addition of 1 mM CaCl_2_ in the GLAC medium recapitulated the emission of luminescence but, as expected, to much higher levels (Figure 4A, open circles). When the Ca^2+^ chelator BAPTA was added to wild-type cells, the luminescence emission was significantly reduced to low levels (Figure 4B, blue triangles), thus indicating that most luminescence signals in CaCl_2_-dependent conditions were due to the presence of Ca^2+^ from external sources. The residual luminescence signal during the chelation of Ca^2+^ by BAPTA was best observed after 60 s of experiment time had elapsed when compared to CaCl_2_ conditions (Figure 4B, blue triangles). This profile is likely to reflect the mobilization of Ca^2+^ from internal sources (for example vacuole, endoplasmic reticulum).

The same experiments were repeated with the *mid1*∆/∆, *cch1*∆/∆ and *cch1*∆/∆/*mid1*∆/∆ mutants. First, when amiodarone was injected to the mutant cells in GLAC medium only, the luminescence recorded was significantly less than that for the wild-type (Figure 4A, closed circles). These data strongly suggest that most of the luminescence signal emitted by the wild-type when incubated under the same conditions was due to external Ca^2+^ sources. The addition of 1 mM CaCl_2_ to the mutants resulted first in a sharp increase in luminescence, which was next interrupted and followed by a second wave of luminescence (Figure 4A, open circles). The addition of BAPTA reduced these effects (Figure 4B, triangles). Of note is that the increase in luminescence was inhibited by a few seconds as compared to CaCl_2_-induced conditions, thus likely linked to the mobilization of internal Ca^2+^ sources as opposed to CaCl_2_-induced conditions. The most dramatic decrease in luminescence was obtained in the *cch1*∆/∆/*mid1*∆/∆ mutant (Figure 4B, orange triangles), highlighting the need to remove both units of this channel complex to maximally inhibit Ca^2+^ influx. Taken together, these results demonstrate the effect of amiodarone on Ca^2+^ flux in *C. albicans*, which acts by perturbing Ca^2+^ homeostasis in *C. albicans*, a step that could eventually lead to cell death.

## 4. Discussion

This study aimed first to establish aequorin as a robust system to probe Ca^2+^ flux in *C. albicans*. Mechanisms that maintain Ca^2+^ homeostasis in this yeast species are important for transition to the hyphal growth stage of this fungus as well as for response to several stresses inflicted by both physical stimuli and chemical insult, such as antifungal agents. 

As mentioned above, aequorin requires coelenterazine as a co-factor and it is only through the recent availability of a water-soluble coelenterazine formulation that sufficient and consistent luminescence signals could be obtained. The water-soluble coelenterazine formulation was also an advantage in vivo when used to detect luciferase-dependent reactions [25]. Other studies performed in filamentous fungi and budding yeast have used the standard form of hydrophobic coelenterazine that is only soluble in organic solvents [19,32]. The reason for the failure of hydrophobic coelenterazine for use in *C. albicans* remains unclear, however cell wall composition and other intrinsic properties of the *C. albicans* cell wall may be involved.

The luminescence signals that were obtained here can be converted and used to quantify the effective free [Ca^2+^] in cells. Using a published formula [33], we estimated basal [Ca^2+^] cytoplasmic concentrations within cells in the range of 0.02–0.05 µM. This is consistent with the concentration range obtained in other fungal cells [19,34]. Alkaline stress as reported here increased this concentration to about 0.75 µM in wild-type cells, and thus is likely to reach toxic levels. As the data of this work show, these high free Ca^2+^ concentrations can, however, be rapidly decreased by the *C. albicans* Ca^2+^ homoeostatic system. It is likely that these high transient Ca^2+^ spikes will in turn activate specific signaling pathways, among which the calcineurin signaling pathway is of significance because it is coupled with stress-response pathways, morphogenesis, and virulence [7,10,35,36,37,38,39].

The cell-based assay used here to report Ca^2+^ fluctuation in *C. albicans* reports the behavior of a whole population. Ca^2+^ homeostasis is, however, highly compartmentalized in cells, and other tools need to be developed in order to obtain a more detailed overview of Ca^2+^ flux within individual cells. One approach is to develop an aequorin detection system that targets specific organelles [40]. A more powerful approach is to perform single cell imaging of Ca^2+^ flux. A FRET (Förster Resonance Energy Transfer)-based Ca^2+^ sensors of the Cameleon family has been successfully used in filamentous fungi to probe Ca^2+^ in individual cells [41]. A recent study reported the use of genetically encoded reporters called GCaMPs in the fungus *Aspergillus fumigatus*. GCaMP reporters are made of a circularly permuted (cp) GFP in between the calmodulin (CaM)—interacting region of chicken myosin light chain kinase (M13) at the N-terminus and a vertebrate CaM at the C-terminus. Binding of Ca^2+^ causes the M13 and CaM domains to interact and the resulting conformational change leads to an increase in cpGFP fluorescence [17]. Future studies should explore the feasibility of GCaMPs in *C. albicans*.

Only a few studies addressing *C. albicans* Ca^2+^ homeostasis are available for comparisons with the current study. It was shown by Wang et al. [26] that alkaline stress could result in a calcium influx from external sources that was dependent on the Mid1/Cch1 channel complex. Cytoplasmic [Ca^2+^] was detected by flow cytometry with a fluorescent dye (fluo-3-AM-ester). The data obtained after alkaline stress resulted in a 120 s time-lapse increase in fluorescence without reversion to baseline fluorescence. This type of profile is essentially different from the current data, in which luminescence adopted a bell-shaped profile with a rapid increase (10 s) to maximal values (Figure 2). This type of response is much closer to a study in which *S. cerevisiae* was investigated for Ca^2+^ flux under alkaline stress [42]. As published by Wang et al. [26] for *C. albicans*, we confirm that alkaline stress response involves the Cch1/Mid1 channel complex. The same complex is also critical for Ca^2+^ homeostasis in many other fungal species [1]. The involvement of *VCX1* in Ca^2+^ homeostasis was addressed by Jia et al. [43]. The authors found abnormal decreases in Ca^2+^ after alkaline stress, which was also observed here. *VCX1*-dependent Ca^2+^ spikes have been reported in *S. cerevisiae* [16]. Taken together, the data collected in this study are consistent with published studies, however our data reveal more detailed dynamics of Ca^2+^ homeostasis for *C. albicans*.

Amiodarone stress followed by Ca^2+^ response has not been yet investigated in *C. albicans*. This drug has a fungicidal effect in pathogens, and it can act synergistically with other antifungal agents [31,44,45,46]. Amiodarone triggers Ca^2+^ influx in *S. cerevisiae*, thus activating at least the Ca^2+^-dependent calcineurin pathway that is necessary to buffer this drug stress [30,45,47,48]. Interestingly, fluconazole was shown to exacerbate the amiodarone-dependent increase in cytosolic [Ca^2+^], thus revealing some mechanistic insight into the synergistic effect between the two drugs. Here, we show that amiodarone stress involves the Ca^2+^ complex containing Mid1 and Cch1 which allows Ca^2+^ uptake into the cytosol from the external environment. Interestingly, the absence of the Cch1/Mid1 channel did not completely eliminate Ca^2+^ uptake by *C. albicans* in the presence of amiodarone (Figure 4A), suggesting the role of additional components (Fig1) [28]. This rapid Ca^2+^ spike is then buffered by Ca^2+^ sequestration systems that likely involve vacuoles and/or the endoplasmic reticulum. More detailed analysis based on the use of specific Ca^2+^ homeostasis mutants could be used to answer this question. Using the strong calcium chelator BAPTA, we also indirectly showed here that amiodarone can mobilize internal Ca^2+^ sources, but the nature of these remains to be established. One of the likely candidates that may increase intracellular calcium from internal stores could be Ycv1, a vacuolar calcium channel important for intracellular Ca^2+^ homeostasis in *C. albicans* [14]. 

## 5. Conclusions

In conclusion, this work established that codon-optimized aequorin can be used in *C. albicans* as a convenient and reliable system with which to probe Ca^2+^ homeostasis. This system can be adapted for use in other pathogenic *Candida* spp. Future work can be undertaken to test not only the importance of specific Ca^2+^ transporters in response to diverse stresses or environmental conditions, but also to address the mechanisms that use Ca^2+^ flux to respond to the challenges presented by antifungal agents.

## Figures and Tables

**Figure 1 jof-07-00319-f001:**
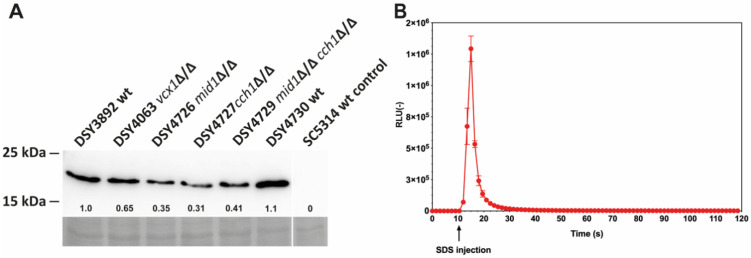
(**A**) Immuno-detection of aequorin in *C. albicans* by Western blot. Protein extracts were obtained as described in the Materials and Methods. Loading control images were obtained by Congo red staining of the nylon membrane after protein transfer. Pre-stained molecular weight standards are indicated at the left side. Designation of *C. albicans* isolates is indicated at the top. Aequorin signals (relative values) were quantified as described in the Materials and Methods using aequorin abundance signals of DSY3892 for reference and are indicated for each protein extract. Details of strain genotypes can be found in Appendix A. wt: wild-type; wt control: untransformed wild-type. (**B**) Total luminescence emission of permeabilized *C. albicans* DSY3892 cells (Appendix A). Cells were prepared as described in the Materials and Methods. SDS injection was performed after 10 s of pre-incubation in GLAC medium (arrow). Results were obtained from duplicate experiments and data were subtracted from the background in cells devoid of aequorin expression (CAF2-1). RLU: relative luminescence unit.

**Figure 2 jof-07-00319-f002:**
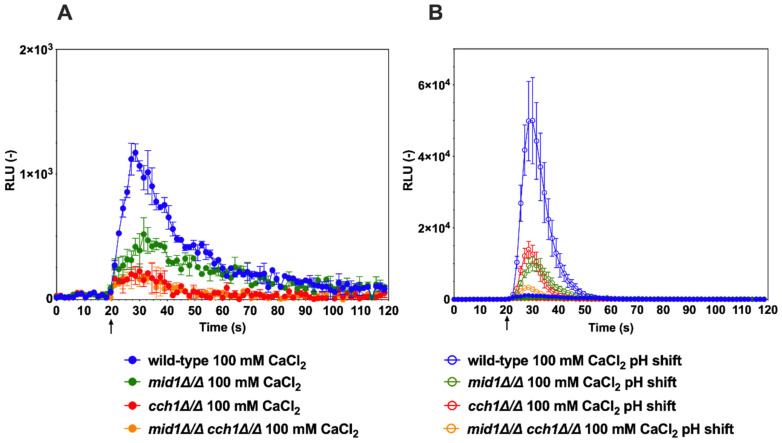
Alkaline stress induces a calcium spike in *C. albicans*. (**A**) Addition of 100 mM CaCl_2_ only. DSY3892 was used as the wild-type control strain. DSY4726, DSY4727 and DSY4729 (*mid1*∆/∆, *cch1*∆/∆ and *mid1*∆/∆ *cch1*∆/∆) mutant cells were prepared as described in the Materials and Methods. Injection of CaCl_2_ (1 M) was performed after 20 s of pre-incubation in GLAC medium (arrow). Results were obtained from duplicate experiments and data were subtracted from background cells devoid of aequorin expression (CAF2-1). Comparisons of RLU values between the CaCl_2_-treated wild-type and mutants were significant (*p* < 0.001) after injections and after up to 50 s of experiment time had elapsed. Comparisons of RLU values between the CaCl_2_-treated mutants were not significant. (**B**) Addition of 100 mM CaCl_2_ and pH shift. Injection of 0.25 M Tris-HCl (pH 7.5) and CaCl_2_ (1 M) were performed after 20 s of pre-incubation in GLAC medium (arrow). Comparisons of RLU values between the CaCl_2_-treated cells and alkaline conditions were significant (*p* < 0.001) after injections and up to 30 to 50 s of experiment time had elapsed. Comparisons of RLU values between the wild-type and mutants under alkaline conditions were significant (*p* < 0.001) and up to 50 s of experiment time had elapsed. Comparisons of RLU values under alkaline conditions between mutants were significant (*p* < 0.001) up to 40 s of time had elapsed between the *mid1*∆/∆ *cch1*∆/∆ mutant and single mutants, although not significant between single mutants. Data in the *y*-axis are shown at a higher scale as compared to panel A.

**Figure 3 jof-07-00319-f003:**
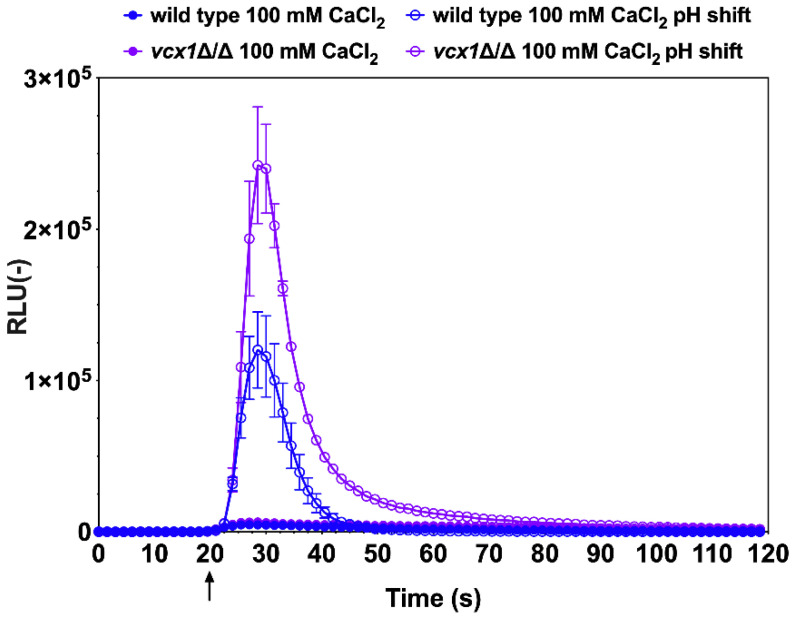
Relevance of the H^+^/Ca^2+^ exchanger, Vcx1, in Ca^2+^ homeostasis in *C. albicans*. Wild-type cells (DSY3892 and DSY4063) and the *vcx1*∆/∆ mutant were prepared as described in the Materials and Methods. Injections of 0.25 M Tris-HCl (pH 7.5) and CaCl_2_ (1 M) were performed after 20 s of pre-incubation in GLAC medium (arrow). Results were obtained from duplicate experiments and data were subtracted from background cells devoid of aequorin expression (CAF2-1). Comparisons of RLU values between the CaCl_2_-treated wild-type and *vcx1*∆/∆ mutants were not significant all time points, although they were significant (*p* < 0.001) in alkaline conditions after injections and up to 80 s of experiment time had elapsed.

**Figure 4 jof-07-00319-f004:**
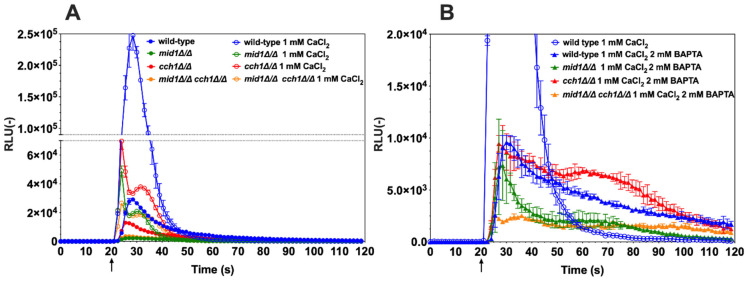
Amiodarone perturbs Ca^2+^ homeostasis in *C. albicans*. Wild-type (DSY4730, DSY4726, DSY4727 and DSY4729), *mid1*∆/∆, *cch1*∆/∆ and *mid1*∆/∆ *cch1*∆/∆ cells were prepared as described in the Materials and Methods. Injection of amiodarone was performed after 20 s of pre-incubation in GLAC medium (arrow). (**A**) Effect of the addition of CaCl_2_ (1 mM). Comparisons of RLU values between the absence and presence of CaCl_2_ in the amiodarone-treated wild-type and mutants were all significant (*p* < 0.001) after injections and up to 45 s of experiment time had elapsed. Comparisons of RLU values in the presence of CaCl_2_ and amiodarone between wild-type and mutants were all significant (*p* < 0.001). Comparisons of RLU values in the presence of CaCl_2_ and amiodarone between mutants were all significant (*p* < 0.001), with the exception between the *mid1*∆/∆ and *mid1*∆/∆ *cch1*∆/∆ mutants. (**B**) Effect of the addition of CaCl_2_ (1 mM) and BAPTA (2 mM), added before the start of the experiment. Results were obtained from duplicate experiments and data were subtracted from background cells devoid of aequorin expression (SC5314). A recapitulation of a CaCl_2_-only condition (panel A) is given for the wild-type comparison. Comparisons of RLU values between the presence of CaCl_2_ and BAPTA chelation in the amiodarone-treated wild-type and mutants were all significant (*p* < 0.001) after injections and up to 45 s of experiment time had elapsed. Comparisons of RLU values in the presence of CaCl_2_ and BAPTA chelation in amiodarone-treated conditions between wild-type and mutants were all significant (*p* < 0.001) with the exception of the comparison between wild-type and the *cch1*∆/∆ mutant. Comparisons of RLU values in the presence of CaCl_2_ and BAPTA chelation in amiodarone-treated conditions between mutants were all significant (*p* < 0.001) but not between the *mid1*∆/∆ *cch1*∆/∆ and *mid1*∆/∆ mutant after 35 s of experiment time laps. Data in the *y*-axis are shown at a lower scale as compared to panel A.

## Data Availability

The datasets generated during and/or analyzed during the current study are available from the corresponding author upon request.

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
