# Peer review of "Aequorin as a Useful Calcium-Sensing Reporter in Candida albicans"

_jof, 2021, doi:10.3390/jof7040319_

Round 1

Reviewer 1 Report

The manuscript 'Aequorin as a useful calcium-sensing reporter in Candida albicans' by D. Sanglard describes the set up of an aequorin-based method for cytosolic Ca2+ detection in Candida albicans. The author utilizes this probe to analyze alkaline stress and amiodarone response in Candida.

The presented results are sound and reproducible, they are clearly described and deeply discussed. Though I think the manuscript has merit and is worthy of consideration for publication in JOF, the author could improve data presentation and discussion to better highlight the impact of the reported results.

MAJOR REMARKS:

  • Fig. 1 and Fig. 2 could be condensed into one figure describing the functionality of the probe.
  • I do not get the point of presenting in the same graph the effect of CaCl2 addition alone or together with pH shift. The graphs are very confused, while the separation of the results would not impinge on the evidence that the signal of alkaline stress is much higher.
  • Figure 5 is even more confused. There is no need to reproduce the data with 1 mM CaCl2 in figure 5B.
  • Introduction and discussion should be improved to better highlight the relevance of Candida as a pathogen and of calcium signaling in its pathogenicity, which is quite superficially cited.
  • The involvement of more transporters in Ca2+ influx from extracellular medium should be discussed, since the addition of 2 mM BAPTA changes the response in mid1 cch1 mutants, suggesting that an influx from the external medium is still present in these mutants. Does the author know of any other transporters in Candida?? What transporters could be involved in Ca2+ release from internal stores? A very obvious candidate could be Yvc1, why did not the author test the involvement of this or other transporters?

Author Response

WE thank reviewer 1 for his constructive comments. Here are our answers (red color)

  • Fig. 1 and Fig. 2 could be condensed into one figure describing the functionality of the probe.

We have now condensed the two Figures into a new Figure 1. Please notice that we added quantification of aequorin in the western-blot analysis as required by Reviewer 2.

  • I do not get the point of presenting in the same graph the effect of CaCl2 addition alone or together with pH shift. The graphs are very confused, while the separation of the results would not impinge on the evidence that the signal of alkaline stress is much higher.

We agree that the data can get confusing when overloaded. We have now removed the ph shifted values from panel B in the new Fig 2. Note that we switched both panels, which also changed the way Figs are described in the text (l 205-210 of revised manuscript).

  • Figure 5 is even more confused. There is no need to reproduce the data with 1 mM CaCl2 in figure 5B.

In this Figure (now Fig 4), we removed the 1 mM CaCl2 data of panel B, with the exception of data for the wild type. We feel still necessary to have these CaCl2 data in this Figure panel to highlight differences in the late time points of the experiment.

  • Introduction and discussion should be improved to better highlight the relevance of Candida as a pathogen and of calcium signaling in its pathogenicity, which is quite superficially cited.

We added several sentences in the Introduction for this purpose (l 41-48 of revised manuscript):

"For example, Brand et al. [4] described that contact sensing and subsequent hyphal development in strongly affected by extracellular calcium chelation.

[Ca2+] flux is also important for C. albicans to cope with different stresses and in particular with antifungal-dependent stresses [6–8]. One of the major stress response pathways in C. albicans is the calcium-dependent calcineurin pathway [9]. Blocking this pathway by chemical or genetic interference leads to a loss of resistance to these stresses [10]. In C. albicans, the calcium-dependent calcineurin pathway is also associated with the maintenance of virulence and thus highlight the importance of [Ca2+] flux control in the context of the host [7]."

and (l. 52-54).

"Interestingly, Yvc1 in C. albicans was shown to play a role in cell polarization and hyphal extension, which is critical for the maintenance of virulence in this pathogenic yeast species [14]."

  • The involvement of more transporters in Ca2+ influx from extracellular medium should be discussed, since the addition of 2 mM BAPTA changes the response in mid1 cch1 mutants, suggesting that an influx from the external medium is still present in these mutants. Does the author know of any other transporters in Candida?? What transporters could be involved in Ca2+release from internal stores? A very obvious candidate could be Yvc1, why did not the author test the involvement of this or other transporters?

We discussed in the Results and discussion section the possible involvement of another calcium channel (for ex. Fig1)

l. 221-226: “Our data also suggest that additional Ca2+ uptake should be present in C. albicans since the absence of the CCH1/MID1 channel complex could not completely abolish the activity of the aequorin reporter system (Fig. 2B). One likely additional candidate in this process could be Fig1, a component of the Low-Affinity Calcium-uptake System (LACS) that is activated under specific conditions in C. albicans [27].”

l. 385-387: “Interestingly, the absence of the Mid1/Cch1 channel did not completely eliminate Ca2+ uptake by C. albicans in the presence of amiodarone (Fig. 4A), suggesting the role of additional components, among which Fig1 [27].”

With regards to the mobilisation of internal Ca stores, we now added a sentence in the Discussion section:

l.392-394: “One of the likely candidates that may increase intracellular calcium from internal stores could be Ycv1, a vacuolar calcium channel important for intracellular Ca2+ homeostasis in C. albicans [48].”

In addition, we corrected small typo errors in the revised version.

Reviewer 2 Report

The manuscript “jof-1110504” authored by Sanglard describes a system based on a codon-optimized aequorin to examine Ca2+ homeostasis in C. albicans. I have some minor comments that could help to improve the manuscript. 1. Appropriate statistical analyses of all data shown in this study should be performed. 2. The author should quantify the immunoblot signals of the Figure 1 and show these data in a graph. How many replicates of the immunoassay were performed? 3. Immunoblot analyses yield data concerning “protein abundance” and not “protein expression”. Please, correct and re-structure statements that incorrectly include this wrong terminology. 4. In the Figure 1 Legend, "western” should be changed to “Western”.

Author Response

We thank the reviewer for his constructive comments. We addressed the issues raised by the reviewer (red color):

1. Appropriate statistical analyses of all data shown in this study should be performed.

We added statistical evaluations in the comparisons of curves obtained in Figs2-4. These descriptions are now to be found in Figure legends. The method used is also added in Materials and Methods (l 167-170 of revised version).

2. The author should quantify the immunoblot signals of the Figure 1 and show these data in a graph. How many replicates of the immunoassay were performed?

We also included the quantification of Western blots in the new Fig 1 of the revised version and did other corrections as suggested by the reviewer. For this, new sentences were added in Material and Methods (l.163-166):

"Quantifications of Western blot signals were carried out by an ImageQuant LAS 4000 System (GE Healthcare Life Sciences, Glattbrugg, Switzerland). Chemoluminescence signals and Congo red staining intensities were quantified by the software ImageQuantTL."

and a comment (l-178-180):

"The expression of aequorin was verified by immuno-blot analysis in wild-type and mutant C. albicans strains used in this study and little variations were observed (Fig. 1A).".

We repeated the immuno-detection with similar results.

also corrected Legend

3. Immunoblot analyses yield data concerning “protein abundance” and not “protein expression”. Please, correct and re-structure statements that incorrectly include this wrong terminology. 4. In the Figure 1 Legend, "western” should be changed to “Western”.

Corrections were made in Figure 1A (notice that Fig 1 and 2 were fused into a new Figure as requested by reviewer 1).

Round 2

Reviewer 1 Report

I thank the authors for addressing all remarks.